# Brain Function, Learning, and Role of Feedback in Complete Paralysis

**DOI:** 10.3390/s24196366

**Published:** 2024-09-30

**Authors:** Stefano Silvoni, Chiara Occhigrossi, Marco Di Giorgi, Dorothée Lulé, Niels Birbaumer

**Affiliations:** 1Institute of Cognitive and Clinical Neuroscience, Central Institute of Mental Health, Medical Faculty Mannheim, Heidelberg University, 68159 Mannheim, Germany; 2ALS Voice gGmbH, 72116 Mössingen, Germany; chiara.occhi@outlook.it (C.O.); marcodigiorgi96@gmail.com (M.D.G.); 3Department of Neurology, Ulm University, 89081 Ulm, Germany; dorothee.lule@uni-ulm.de; 4Institute of Medical Psychology and Behavioral Neurobiology, Tübingen University, 72076 Tübingen, Germany; niels.birbaumer@uni-tuebingen.de; 5Physical Medicine and Rehabilitation, University of Padua, 35121 Padua, Italy

**Keywords:** completely locked-in state, operant learning, neurofeedback, reinforcement, goal-directed thinking, cognition, psychosocial context, communicative barrier

## Abstract

The determinants and driving forces of communication abilities in the locked-in state are poorly understood so far. Results from an experimental–clinical study on a completely paralyzed person involved in communication sessions after the implantation of a microelectrode array were retrospectively analyzed. The aim was to focus on the prerequisites and determinants for learning to control a brain–computer interface for communication in paralysis. A comparative examination of the communication results with the current literature was carried out in light of an ideomotor theory of thinking. We speculate that novel skill learning took place and that several aspects of the wording of sentences during the communication sessions reflect preserved cognitive and conscious processing. We also present some speculations on the operant learning procedure used for communication, which argues for the reformulation of the previously postulated hypothesis of the extinction of response planning and goal-directed ideas in the completely locked-in state. We highlight the importance of feedback and reinforcement in the thought–action–consequence associative chain necessary to maintain purposeful communication. Finally, we underline the necessity to consider the psychosocial context of patients and the duration of complete immobilization as determinants of the ‘extinction of thinking’ theory and to identify the actual barriers preventing communication in these patients.

## 1. Ideomotor Theory of Thinking

The concept of ideomotor thinking, already expressed long ago by Aristotle (384–322 BC) and followed up by William James (1890) and Carl Georg Lange (1887), states that the perception of movements creates the thinking of that particular movement and therefore thoughts are the consequences of movements and not vice versa [1,2,3]. A modern conceptualization of this thought–action–consequence construct is represented in Figure 1 [4]. The response (R) caused by a particular stimulus (S) produces some effect (E). The discrepancy between the anticipated and actual effect produces a modification in the response planning to achieve the target (desired) effect [5].

In the context of the “ideomotor theory of thinking”, the neural representation of an idea is produced once an external discriminative stimulus triggers an S–R–E contingent sequence through instrumental (voluntary) learning. This ideomotor concept is supported by the evidence that the motor cortex is involved in the covert stages of an action, providing a person with information on the feasibility and the meaning of anticipated actions [6], although other brain areas might be involved in these processes. As previously postulated, the lack of desired effects and reward contingency in complete paralysis, such as in the late stage of amyotrophic lateral sclerosis (ALS), would affect the production of ideas and affect anticipating thinking. As a consequence, it would progressively lead to the extinction of response planning and goal-directed ideas after a long period of complete immobility, with no chance to produce a desired effect [5,7]. This hypothesis called the “extinction of goal-directed thinking” may be the cause or consequence of progressive reduction in synaptic plasticity and neuro-modulatory control in specific brain areas. In the first version of this hypothesis, the length of the period of complete paralysis was not clearly defined, leaving a certain degree of uncertainty for the effect of complete extinction. Several discussions on this hypothesis have suggested that learning brain–computer interface (BCI) control or any other S–R–E contingency *before* the onset of complete paralysis should prevent the extinction, presumably enabling the transfer of skills from the locked-in state to the completely locked-in state [5,7]. However, the psychosocial context and the history of complex communication needs in (ALS) long survivors [8,9,10] have not been thoroughly investigated and discussed in-depth to date (see the review by Birbaumer, 2024) [11]. That is, the paucity of the BCI studies on completely paralyzed patients renders it difficult to verify the above hypothesis, especially for those studies that have implemented very short-term tests [12] and, to the best of our knowledge, have not been replicated to date. In addition, there exist single-case invasive BCI studies reporting very accurate speech recognition when the implanted patients are engaged in an attempt to perform (or in covert) speech tasks (e.g., [13]). However, if remnant voluntary motor control has been assessed in the patient, the study cannot be used to verify the ‘extinction of thinking’ hypothesis as it does not represent a completely locked-in case. For instance, in the work by Card and colleagues (2024), the patient retained eye and neck movements at the time of the enrolment [13].

The above hypothesis of the ‘extinction of thinking’ would be disproved by a single case of a completely locked-in patient able to communicate with a BCI-system based on instrumental (voluntary) learning *after* entering the state of complete paralysis and without a preceding history of control over the same BCI system, i.e., a patient who enters the completely locked-in state *before* learning BCI use [5,7]. That is, a single case-report such as that would suffice to contradict the ‘extinction’ hypothesis. Therefore, it turns out that *learning* is one of the essential conditions to be examined in-depth. We hereby introduce the case of a completely paralyzed person, who was able to effectively use a brain–communication system based on neurofeedback principles [14]. A thorough retrospective analysis of the communication strategies and outcomes (i.e., the sentences produced by the patient) will be presented in next paragraphs, in light of the mentioned S–R–E model, the operant conditioning principles and the postulated “extinction of goal-directed thinking” hypothesis. The data presented herein pertain to the communication sessions of the patient as they have been reported by Chaudhary and colleagues (2022) [14]. The observations touch on different aspects of cognition and learning to provide the reader with a general overview of the residual preserved capabilities in this patient.

## 2. Communication in (Completely) Locked-In State

### 2.1. Clinical and Communication History of the Patient

The patient considered herein was finally diagnosed with ALS in August 2015. He used an eye-tracking assistive device (Tobii, MyTobii, Danderyd Stockholm) for communication purposes from August 2016 until August 2017. Since then, the inability to fixate his gaze prevented the use of the eye-tracker for communication. In the subsequent period, the family adopted a manual spelling system to communicate by observing his eye movements from behind a transparent board with printed color letters. From March 2018 to February 2019, a non-invasive communication solution based on an electro-oculogram was successfully implemented, although the ability to communicate deteriorated progressively due to the decline in the neuro-muscular control of the eyes [15]. At the time of the non-invasive communication study, the functional score of the patient was 0 (see ALS Functional Rating Scale—Revised) [16]. The last successful communication through the non-invasive solution occurred in February 2019. The patient was then implanted in March 2019. Two 64 microelectrode arrays were implanted in the dominant left supplementary and primary motor cortex (8 × 8 electrodes each, 1.5 mm length, 0.4 mm electrode pitch; Blackrock Microsystems LLC). A magnetic resonance imaging scan of the head was performed to plan and support the microelectrode array placements during the neurosurgery intervention. The implantation was carried out using a neuro-navigation system, anatomical landmarks, and a pneumatic inserter. The connector of the microelectrode arrays was fixed to the calvaria and exited through the skin (the interested reader is referred to reference [14] for details). The patient could not reliably communicate until the first successful neurofeedback attempt about four months after the implantation, i.e., June 2019. The reported period of neurofeedback conversations was recorded until June 2020.

As reported in the original article, the procedure for the neurosurgical intervention “was approved by the Bundesinstitut für Arzneimittel und Medizinprodukte (“BfArM”, The German Federal Institute for Drugs and Medical Devices). The study was declared as a Single Case Study and received a special authorization (“Sonderzulassung”) by BfArM, according to §11 of the German Medical Device Law (“Medizin-Produkte-Gesetz”) on December 20, 2018, with Case Nr. 5640-S-036/18. The Ethical Committee of the Medical Faculty of the Technische Universität München Rechts der Isar provided support to the study on 19 January 2019, along with the explicit permission to publish on 17 February 2020.” [14] (p. 7).

In the following, we will consider only the communication sessions reported by Chaudhary et al. (2022) [14]. No new or additional data were collected in the present retrospective study.

### 2.2. Communication Devices

The data acquisition devices are listed in Table 1 (see second column). Their usage depended on the stage of the disease and the residual capabilities of the patient. With the exception of the second communication period, in which the eye movement observation spelling procedure was adopted, without digital recording, three different communication devices were employed.

In the first period, an infrared light eye-tracking device (Tobii, MyTobii, Danderyd Stockholm) was used to track the pupils. The movements of the pupils were translated to control the pointer in the computer screen. The eye-tracker latency was <70 ms. A character/symbol was selected by fixing the gaze over the corresponding location for a customized interval. In the third period, a neurophysiological signal amplifier (V-Amp DC, Brain Products, Gilching Germany) was used to record the electro-oculogram during the auditory presentation of characters/symbols. Four Ag/AgCl active electrodes served to record the horizontal and vertical eye movement. The sampling rate was 500 Hz. Each signal was bandpass filtered (0.1–35 Hz). Not moving the eye and horizontal eye movement were coded as “no” and “yes”, respectively, allowing the patient to skip or select the presented item [15]. In the fourth period, the neuro-electric signal processor device (CerePlex E and NSP, Blackrock Microsystems LLC) was employed for the invasive recording and processing of the signal from the array placed in the supplementary motor area (SMA). The 64 microelectrodes were sampled at 30 kHz and bandpass filtered (250–7500 Hz). The spike rate metric of the neural response served as feedback. The signal was presented as tones changing in frequency. The modulation towards the low or high tones indicated a negative or an affirmative response, respectively. The interested reader is referred to reference [14] for details.

### 2.3. Communication-Oriented Strategies

Considering the period from August 2016 to June 2020 from an (operant) learning perspective, the patient learned the following four different tasks to communicate: usage of the eye-tracker, eye movements associated to a spelling system, eye movements associated to a digital auditory speller, and the neurofeedback task, which consisted of presenting the patient with increased and decreased firing rates of cellular responses from the array in the supplementary motor area. In the neurofeedback task, this feedback information to select letters and form words from his auditory speller was fed back via the auditory system as a tone changing in frequency [14]. The four communication strategies are presented in Table 1. The communication strategy of the first three periods involved the control of eye movements. While during the last period, in which the neurofeedback task was employed, the mental strategy (or ideas) used to intentionally modulate the tones is unknown (i.e., not reported), nevertheless, comparing the four periods, we can observe substantial differences concerning the task. It is also worthwhile to note that, in the third and fourth periods (characterized by the electro-oculogram-based system and the neurofeedback task, respectively), the digital auditory scheme was the same; however, the task, the discriminative stimulus, and the response–effect time interval were not. The discriminative stimuli of the third and fourth periods were a “no”/“yes” question with a known answer and a target tone, respectively. Moreover, the response–effect time interval was much shorter in the neurofeedback task (i.e., 250 ms), and the feedback tone was varying in the frequency range of 120–480 Hz, where high tones indicated an affirmative response.

From a theoretical point of view, each of the four applied communication strategies entails the shaping of a response aimed at strengthening a specific behavior [17]. Examples of learning and beneficial effects of shaping have been already reported in a range of clinical applications [18,19,20,21]. In other words, throughout the training, the strengthening of behavior through operant reinforcement enables the acquisition of a new skill [22]. Thus, the neurofeedback task was a new task for the patient. This holds true despite the fact that we cannot rule out that the patient used the imagery of eye movements to modulate the firing rate and tones. Additionally, we believe that the training carried out from day 86 to day 98 after implantation was essential in establishing the first successful S–R–E associative chains and learning of the matching of tones strategy [14]. Following this strategy, generalization occurred, and the selection of characters and wording of sentences was possible and accurately documented from day 106 after implantation.

### 2.4. Reliability of the Communication Interfaces

In the real environment scenario presented herein (home of the patient), the reliability of the communication interfaces is considered only qualitatively because the focus is centered on crucial aspects such as learning and the final outcome of the communication sessions—the intelligible sentences produced by the patient during free-spelling. In other words, we were particularly interested in sessions characterized by fluent and effective communication, which were able to trigger an interaction of the patient with his family, caregivers, and the experimenters.

The performance and reliability of sessions with no intelligible outcome are diversely informative in this context because the learning processes may have been altered or prevented due to different factors (e.g., low vigilance level, drowsiness, shortened attention span, fatigue). Therefore, they should be used thoughtfully to argue learning effects, which are essential determinants of the ‘ideomotor’ and ‘extinction’ theories.

Following the communication history of the patient, in the first period, we assumed that the patient could have benefited from the usage of the eye-tracker, although no data are available for this period of about a year. In ALS patients, the average typing speed using an eye-tracking device in a copy-spelling task is greater than 23 characters per minute, with an average error rate of 1% [23]. For the second period, in which a manual spelling procedure was used, no performance data are available. However, it is known that the family used this procedure for seven months. In the third period, the reliability of the communication interface was evaluated by the correct copy-spelling and free-spelling of meaningful sentences. In copy-spelling, 7 out of 14 sessions were correct, while for free-spelling, a meaningful sentence was produced in 5 out of 9 sessions [15]. For the last period, intelligible output (rated independently by three observers) from the invasive speller interface was produced by the patient on 44 out of 107 days in which the free-spelling sessions were enabled [14].

The overall picture of the invasive communication interface’s reliability tells us that, as the disease progressed, the capability of the patient to interact via free-spelling decreased; however, this capability was not completely hindered.

## 3. Brain Function and Cognition

As a premise, we will consider the facts, events, persons, situations, and circumstances emerging from the conversations between the patient and the caregivers as they were reported by Chaudhary and colleagues (2022) [14]. Only a structural magnetic resonance imaging was performed prior to the neuro-surgical intervention, which did not reveal any significant structural abnormality, nor brain atrophy, or signs of major neural degeneration. In short, the following arguments are derived from behavior, i.e., the intelligible sentences produced by the patient and their meaning from a neuroscientific viewpoint.

### 3.1. Comprehension of Instructions and Expressive Components

After surgery, the patient was asked to pursue the previously effective communication strategy using eye movement imagery, which yielded only chance-level results [14] according to the metric published by Müller-Putz and colleagues (2008) [24]. Other brain communication strategies were tested, such as the imagery of passive movements and motor imagery of hand movements, during the four months after the implantation; however, they were unsuccessful. Only when NB (i.e., last author of this study) proposed the neurofeedback strategy, the firing rate of neuronal responses (fed back as tones) allowed the patient to control the speller. Therefore, the patient demonstrated the ability to listen to and understand the instructions of the task to be accomplished. Due to partially compromised vision [15], the instructions were provided auditorily. The neurofeedback task involved the frequency modulation of a tone. These abilities entail almost perfect hearing ability and language comprehension.

Additionally, on some occasions, the patient demonstrated the capacity to perform goal-directed thinking, abstraction, and well-constructed elaboration of his own communication experience since he provided the experimenters with direct feedback (see Table 2, section Social). For instance, on day 107 after implantation, he expressed gratitude to ‘Birbäumchen’ (the nickname given to the mentor of this invasive communication initiative). On day 295, he demonstrated purposeful reasoning, suggesting specific changes to the speller interface (‘tell alessandro I need to save edit and delete whole phrases and all of that into the list where [patient’s son name]’), and on day 247, he provided the experimenters with reward (‘boys, it works so effortlessly’). This would indicate that written language production (more precisely mental spelling) and high-level cognitive functions were working in synergy, with the aim to optimize the actual communication outcome [25] (pp. 1–16). At that point, the patient was already completely locked-in for more than a year.

Although a structured and standardized linguistic and cognitive assessment was not reported in this patient, we can consider that the above-mentioned abilities require excellent linguistic verbal comprehension (on a lexical/semantic and morpho-syntactic level) and preserved executive functions (especially divided attention, alternating attention, working memory and inhibitory control). Hence, we speculate that intact processing occurred in most cortical regions, including the auditory cortex and Broca’s and Wernicke’s areas in the dominant frontal and temporal lobe, respectively, and the arcuate fasciculus, during each successful communication session [26] (pp. 398–423), [27,28].

### 3.2. Memory

The patient showed preserved long-term memory capabilities because the digital auditory speller scheme (see Figure 2), previously introduced in the non-invasive communication solution, was maintained after implantation. The usage of the same speller scheme likely facilitated its retention. The patient maintained a mental map of five subsets of characters encoded by colors (i.e., yellow, green, red, blue, and white). The colors presented by the interface via auditory verbal production were part of the discriminative stimuli used in the instrumental spelling procedure. For instance, he could remember the association of the subset of letters {*E*, *A*, *D*, *C*, *B*, *F*} with the ‘yellow’ color (see Figure 2, first row). Retrieval of the map and comparison with the discriminative stimuli presented during spelling, most likely engaged long-term memory processes, working memory, and executive functions. Therefore, we speculate that a certain degree of integrity of medial temporal lobe structures (among others, entorhinal cortex, perirhinal cortex, hippocampus, and para-hippocampus), frontal lobe, and pre-frontal cortex supported these aspects of free-spelling [26] (pp. 344–360), [29].

Clear evidence of preserved episodic memory function can be observed in at least three sentences where the patient referred to ‘episodes’ that occurred in the past (see Table 2, section Memory). As reported above, on day 107, the patient spelled (i.e., retrieved and selected) the first name and nickname of the mentor to show appreciation, retrieving past information from a few years earlier when he was introduced to one of the authors (NB). A second sentence concerned an event during which he listened to music (‘the album by Tool’) and, on day 245, he expressed the wish to listen to the same music again aloud. The third one was a previous conversation (or discussion) about an event planned by the family (‘My biggest wish is a new bed and that tomorrow I come with you for barbecue’) and, on day 462, he manifested the wish to participate in the ‘barbecue’, providing temporal information of the upcoming event. Food-related wishes (see Table 2, section Social) may also have been linked to the history of previous meals, plausibly indicating the episodic retention of past events. It is therefore reasonable to conclude that these statements were improbable without the intact functioning of the medial temporal lobe.

### 3.3. Emotional State

On many different occasions, the patient demonstrated the need to express feelings and social interactions (see Table 2, section Social). For instance, sentences such as ‘I love my cool son’, ‘and now a beer’, ‘come tonight [to continue with the speller]’, and ‘my biggest wish is a new bed and that tomorrow I come with you for barbecue’, show the interest to maintain inter-relationships with relatives and caregivers. Almost all the expressed wishes, if not all, appear to denote the tendency to anticipate positive or beneficial effects (i.e., happiness, well-being, pleasure, satisfaction). In the advanced stages of ALS, there is evidence that the perceived quality of life is high [30,31]. This is likely due to effective coping strategies in these patients [32], where high satisfaction with their own life, high social orientation, strong emotionality, and low aggressiveness are thought to underlie their emotional adjustment [33]. In these patients, the self-perceived psychosocial state significantly deviates (towards better outcomes) from the negative judgement of caregivers and age-matched healthy persons [34]. Therefore, it might be argued that the patient’s anticipation of positive or beneficial effects is indirect evidence of a good quality of life in general [10] and of volitional intentions linked to expected perceptual changes [35], mediated by the caregivers.

An explanation accounting for emotion-regulatory processes in life-threatening conditions will be discussed later, following an overview of the role of feedback and reinforcement. We emphasize here that emotion-sensitive brain areas such as the anterior cingulate cortex [36], amygdala, ventromedial pre-frontal cortex [37,38], and insula [39] have been found to be responsible for emotional regulation and emotional learning processes. Thus, we argue that a certain integrity of these neural structures was still in place at the time of the free-spelling sessions.

### 3.4. Sensory and Proprioceptive Processing

As explained above, the auditory system of the patient played a crucial role in the general understanding of instructions, language comprehension, the neurofeedback task itself, and, reasonably, verbal feedback from the caregivers. Regarding vision, which was partially compromised due to a probable degenerative process of the cornea [40], in one sentence on day 254 (see Table 2, section Sensory), the patient referred to the need to keep eyes lubricated (‘everybody must use gel on my eye more often’). Although he expressed the desire to watch movies or cartoons (on three occasions on days 253, 309, and 461), it is difficult to argue whether the use of gel was directly associated to vision itself (i.e., to maintain the residual integrity of visual ability) or to the necessity to get rid of an itch or similar sensations. Therefore, it is difficult to argue for the preserved processing of the visual cortex.

In three different situations, the patient referred to the need of wearing socks for the night (‘no shirts but socks [for the night]’ on day 244), the desire for a head massage by his mother (‘Mom head massage’ on day 247), and the desire for a new bed (‘my biggest wish is a new bed and that tomorrow I come with you for barbecue’, on day 462). We argue that these three statements were based on the need/desire to feel warmth and to obtain a beneficial effect from skin-to-tissue (socks) and skin-to-skin (massage) contact and a comfortable position at bed. In short, a combination of physical and psychosocial well-being. It is therefore reasonable to state that somatosensation and touch processing were partially preserved during these events. In turn, that implies the partial preservation of the somatosensory cortex [26] (pp. 398–423), as already suggested in a completely locked-in patient by Ramos-Murguialday and colleagues (2011) [40].

The above argument is also substantiated by evidence of preserved proprioceptive processing on at least four occasions (see Table 2, section Proprioception). This ability refers to the awareness of one’s body position in space. We limit this speculation to the patient’s awareness of the position of the head and of one of his hands, although it might be extended to the entire body (given the expression of the necessity of a new bed). The patient provided the caregivers with instructions on the position of the head three times (on days 161, 251, and 461). In particular, on day 461, the instructions were likely related to a social interaction (‘when visitors are here, head position always very high’). In one circumstance, the instructions were related to the position of his hand (‘everybody should put my hand direct on my stomach’, on day 344).

Proprioception is known to depend on the primary somatosensory cortex, which receives projections from the thalamus [41]. The increasing integration of complex tactile peripheral responses, including those from slowly adapting mechanoreceptors (i.e., Merkel’s disks and Ruffini’s corpuscles, [42]), thermoreceptors, and nociceptors, occur in the more posterior brain regions, such as the lateral parietal cortex. While affective touch processing is thought to be associated with the insula [41], vestibular information processing is thought to take place in the inferior parietal lobule, the primary vestibular area, and the superior temporal gyrus [26] (pp. 398–423). It is reasonable to conclude that integrity of these brain regions was preserved at the time of the above-mentioned conversations.

## 4. Learning and Reinforcement

We now step back into the instrumental learning procedure to focus on the central aspects of the shaping, namely reinforcement.

The patient successfully learned to modulate his own cortical activity to match the frequency of the neurofeedback tone to the target [14]. He also used this skill for free-spelling. From a neuro-functional point of view, learning and the internalization of a goal after the first S–R–E associative contingencies need intact processing within the cortico–thalamic–striatal loop [43,44] and the preserved plasticity of the N-methyl-D-aspartate (NMDA) receptor-dependent synaptic connections of this system [45]. It has been proposed that the NMDA synapses become strengthened through learning as the post-synaptic membrane potential should vary depending on the prior cell activity history [43]. In addition, the reticular nucleus of the thalamus is thought to act as a topographically specific inhibitory feedback circuit, gating (i.e., mediating) the ascending thalamic information to the cortex and thus operating as a reward-regulatory neural component [43]. These two prerequisites (intact cortico–thalamic–striatal processing and NMDA plasticity) for abstract skill learning were tested in a BCI task in healthy controls and epileptic patients [44,46], consisting of the modulation of slow cortical potentials, and for learning BCI neuro-control in animals [45]. Although, in both these studies, the response rewarded originated from the somato-motor cortex, it is interesting that the reported subjective mental strategies in the former experiment spanned very different (more or less) abstract ideas, such as sensations, feelings, actions, and emotions [44]. Different imagination strategies to modulate the alpha-rhythm activity, recorded non-invasively with scalp electroencephalography, were also reported by the participants of the first-ever neurofeedback experiment in 1962 [47].

In the patient considered herein, the source of the feedback originated from a few channels of the multi-electrode array implanted in the left supplementary motor area, and the feedback was provided through the auditory system [14]. There exists evidence of cortico–striatal connections involving SMA and pre-SMA, which are in turn also connected to the Broca’s area in the dominant hemisphere [48,49]. The pre-supplementary and supplementary motor areas are thought to support sound, speech, and auditory imagery processing, in addition to guided auditory perception [49]. This would suggest a candidate distributed neural substrate (encompassing the striatum, the auditory cortex, the supplementary motor area with their interconnections), supporting procedural learning implemented through the neurofeedback task.

### 4.1. Perception of Reinforcement

Within each S-R-E sequence, the perception—referred here as audible sound—of the mental strategy used by the patient to match the target tone consisted of two components, the feedback and the reinforcement. The former was used alone in the training sessions named “feedback without reward”. The latter was used in combination with the former in the training sessions called “feedback with reward”. The physical feedback consisted of the modulation of the pitch frequency according to the spike rate metric, reflecting the SMA cortical activity, until the target tone (“no”/“down” or “yes”/“up”) was reached. This was one of the core components of the operant learning procedure. First, the perception of this auditory information was easy. Second, the access was almost immediate due to the short response–effect time interval (every 250 ms, the frequency of the pitch was updated). This is consistent with the notion that conscious decision and perception of “will” depends on the close contiguity in time between the decision and the response [7]. Third, the gradual modulation of the frequency of the tone constituted an immediate and accessible metric of the performance of the task.

On each day, once the training session “feedback without reward” was complete, the reinforcement was introduced in the session “feedback with reward” as a sound (lasting 250 ms), indicating successful matching of the tone. This sound constituted an additional component to strengthen the effect of the S–R–E associative chains and thus to refine the specific internalization of the response used to accomplish the task. The meaning of this sound was simple, such as “Great! You did the job!”. Additionally, the experimenters chose an accuracy threshold of 80% in the “feedback with reward” sessions to enable subsequent spelling sessions. Over the reported period (from June 2019 to June 2020), there were 281 “feedback with reward” sessions preceding the free-spelling sessions. Out of the 5700 trials, 4936 were correct (86.6%) [14]. Then, the basic procedural skill (matching of tones) was employed to select characters through the auditory speller interface.

The implementation of these three levels of complexity, namely “feedback without reward”, “feedback with reward”, and “free-spelling”, constituted a kind of shaping procedure (see Figure 3), although not in the strict sense, enabling slow but gradual improvement [21,50,51,52]. The daily increase in complexity was gradual to give the patient time to consolidate the responses. A continuous reinforcement schedule was used in the operant learning procedure, encompassing positive reinforcement [22]. The feedback and operant reinforcement provided immediate information about the performance of the task [50]. There was no punishment. Frustration was also avoided, enabling spelling only with a positive reinforcement rate of 80% [53,54]. We argue that all these elements facilitated the use of implicit skill learning resources to accomplish the task [55]. Multiple (repetitive) executions of the neurofeedback task and the short response–effect time interval (i.e., 250 ms) to provide the physical feedback (in the frequency range 120–480 Hz) and the physical reinforcement should have induced long-term potentiation of NMDA synaptic efficacy [56] within the above-mentioned cortico–thalamic–striatal loop [43], encompassing the left supplementary motor area and the auditory cortex, with the final effect of mastering the modulation of the tones.

### 4.2. Positive Reinforcement

The auditory speller interface reasonably enabled a different form of reinforcement, as communication was the main objective of the invasive approach, given the inefficacy of other communication strategies in the complete locked-in state. According to the social learning theory, “most human behavior is not controlled by immediate external reinforcement” and “actions are [therefore] regulated to a large extent by anticipated consequences” [57] (p. 3). The patient was able to express wills, needs, desires, and emotions as described above. One might wonder whether these wishes, needs, and requests were fulfilled and thus meant positive reinforcement to the patient. Did they produce some effect? Were these anticipated consequences fulfilled? Theoretically, we may reasonably assume that the communication act itself, when successful, produced a positive reinforcement in the form of, for instance, comments, thanks, apologies, statements of emotions, and gestural behavior from caregivers. In other words, this type of reinforcement may have strengthened the awareness of having succeeded in expressing a message. Additionally, we may assume that the caregivers satisfied the expressed needs and desires like all intentions and purposes. All these actions or consequences can be interpreted as a kind of emotional reinforcement, a form of reward involving the affective sphere, although we cannot measure its valence objectively. We can only hypothesize that every effect (E), mediated by the caregivers, may have contributed to maintaining a good standard of well-being.

While the immediate consequence of the physical—produced by sound waves—feedback and reinforcement was to enable the conveying of symbols, such as “no” and “yes” (meaningless per se if not contextualized), the auditory speller interface allowed for the presentation of meaningful sequences of symbols related to the (subjective) experience of the patient, such as, for instance, the statement “no shirts”/“yes socks” expressed in the sentence ‘no shirts but socks [for the night]’ on day 244 [58] (pp. 77–91). That is, the speller was the carrier of meaningful sequences of symbols. In turn, the consequence of the spelled request of the patient produced emotional reinforcement from the caregivers. Hence, the invasive communication system enabled a bidirectional interaction between the patient and his caregivers. The selected words were functional to the social exchange [59] and, most probably, to the emotional (affective) reinforcement. However, in complete paralysis, this form of emotional loop cannot be closed if the basic physical S–R–E associative chain does not take effect. In short, in the completely locked-in state, it occurs only if learning takes place.

### 4.3. A Modified Learning Model of the Stimulus-Response-Effect Unit

As explained above, the repetitive completion of several (physical) stimulus–response–effect contingencies presumably led the patient to receive the emotional reinforcement upon the correct spelling of his wishes and the possibility for the caregivers to fulfil them. Given that this form of reinforcement might have produced physical, emotional, and psychosocial effects, we may assume that the same S–R–E model can be used for both forms of effects, physical and emotional, with an impact on the psychophysical and psychosocial state. We propose a change in the S–R–E model represented in Figure 1 (see dashed line), reflecting the fact that the reinforcement has induced a change in the state of the patient.

In (completely) locked-in persons, the capability to induce changes in their state is primarily mediated by the brain-based speller and reinforcement by caregivers. The speller assumes a crucial function to express health-related issues such as the experience of pain, distress, headache, and anxiety, among many others. In these conditions, the pain/distress perception (stimulus), the communication of such feelings by means of a brain-based speller (defensive response), and the consequences that the social environment is able to provide (effect) can be thought of as an entire unity with beneficial effects on the quality of life and emotional state. In theory, this ultimate objective is based on the basic human right to communicate, written in many declarations including the International Classification of Functioning, Disability, and Health [59,60], which aims to increase the opportunities for social participation by removing (communicative) barriers. In addition, it is strongly recommended in the European guidelines on the management of amyotrophic lateral sclerosis [61].

### 4.4. Ideomotor or Ideosensorimotor Theory of Thinking?

Aristotle’s view and successive theories supporting the thought–action–consequence construct suggest that all thoughts (ideas) originate from movement patterns [62] and the perception of peripheral muscular reactions [2]. The place or, better, the brain region(s) where this representation of the thought of an action takes shape is a matter of debate. Although it is generally known that covert stages of an action (such as imagination and learning by observation, among others) activate the motor cortex [6], it does not indicate that the motor system itself exclusively accounts for movement imagination. As described above, through the cortico–thalamic–striatal loop connections and interplays with the auditory system [48,49], other complementary candidates for the representation of ideas are the pre-supplementary and the supplementary motor cortex, and the auditory and, presumably, the somatosensory cortices. This is supported by the fact that learning occurred in complete paralysis (i.e., complete absence of voluntary movements), with an intense engagement of the auditory sensory system, language-related brain areas, and the feedback over the dominant left supplementary motor area. The assumption of an extended neuronal substrate functional to the ideomotor mechanism is further corroborated by recent hypotheses, one of which proposes that the anterior midcingulate cortex plays a central role [63]. In complete paralysis, the “ideomotor” theory of thinking might be called the “ideosensorimotor” theory of thinking.

Independently of the direction of the cause–effect chain between “idea” and “action”, we emphasize here that volitional, linguistic, and conscious thinking forms a unity of action with its consequences. In completely locked-in patients with previous associative experiences between intentions and actions, hence, with learning and memories of such experiences, this building block forms new stimulus–response–effect associations, even with non-motor responses [11].

This specific change in the construct (from “ideomotor” to “ideosensorimotor” theory of thinking) might open and stimulate new research approaches to the communication problems in the completely locked-in state. For instance, one of the currently less investigated domains for brain–computer-based communication is represented by touch, vibration sense, and proprioception [64], especially in pathological conditions [65]. Soft touch, slip on the skin, texture stimulation, or low/fast vibration and thus somatosensory and proprioceptive processing could represent targets for neurofeedback aimed at mastering cortical modulation. As for the auditory cortex, the somatosensory system processing of touch and/or vibration might provide easy, fast, and immediate access to physical information of feedback and reinforcement. It has been proposed that muscle mechanoreceptors and joint receptors are less affected by the long-term immobilization [40]. Therefore, preserved joint receptors such as Ruffini endings, Pacinian endings, and Golgi tendons [66] could represent targets for the physical feedback and reinforcement. However, this approach needs to be coupled with auditory (or visual) information to enable the selection of symbols (“no”/“yes”, or alphanumeric characters) and convey their semantics.

### 4.5. Determinants of the ‘Extinction of Thinking’ Concept

We now step back to the postulated hypothesis of the extinction of response planning and goal-directed ideas because all the presented arguments call for the reformulation of this theory.

If we try to examine the reasons for the success of the described communication experience, many factors may account for the positive skill learning acquired through auditory neurofeedback. No significant structural abnormality was found in the brain of the patient before the brain surgery. The period of complete paralysis before the first successful neurofeedback attempt was relatively short (about four months). The auditory speller interface was already in use before entering the completely locked-in condition, although the neurofeedback task was new. Before entering the completely locked-in state, the patient expressed his will and gave informed consent to the neuro-surgical intervention using eye movements for confirmation. The quality and sampling rate of the neuro-electric signals recorded from the implanted arrays was higher compared to the non-invasive recording techniques.

Moreover, with regard to emotional processing, there exists evidence, especially in older adults and in persons with difficult life conditions, of enhanced sensitivity to positive than negative information. This phenomenon is known as the *positivity effect* and is thought to influence, among other domains, attention and memory [38,67,68]. Factors such as the optimization of emotional satisfaction [69] and enhanced tailoring of emotion-regulatory strategies to contextual demands with age [70] are thought to lead to positivity preferences. These arguments, substantiated by behavioral and neurobiological findings [38,71], might explain the affective state of the patient considered here and his predisposition to this particular positive communication experience. This is also in line with the finding that perceived affect regulation appears to be one central component of resilience in old age, threatening conditions, or difficult situations [72]. We can add, although it was never explicitly reported, that the patient could benefit from appropriate and good psychological, psychosocial, and technological support from the family, the caregivers, and the experimenters. This social support is fundamental [60]; if the caregivers/researchers do not become a friendly and trusted part of the family, forming a close positive relationship with intimate cooperation over long periods of time in the homes of the patients, the success of brain-based communication is unthinkable [11]. An “objective”, neutral, and “scientific” attitude in such a situation is counterproductive.

It should be considered that a consistent proportion of ALS patients do not consider tracheostomy [73,74], with death as the consequence, while a portion of patients in this condition (artificial ventilation and feeding), with motor inability and preserved cognitive processing, are willing to begin and pursue the initiative (non-invasive or invasive) to enable communication. Other ALS patients in the same condition opt for the discontinuation of mechanical ventilation [75], particularly with the lack of social support. This might be partially related to the psychosocial context in which they live and the respective societies legal and political regulations toward hastened death, and partially to the response that society and the research community are able to provide to guarantee them effective care, including tracheostomy, their right to communicate, and the facilitation of 24 h 7-day care. For instance, in the United States, up to 95% of ALS patients do not opt for tracheostomy [76] and die. In Europe, the situation is similar [74], although not as extreme [77,78], except in countries with “liberal” euthanasia laws. In addition, there exist different orientations to disease management, with patients seeking and patients unwilling to consider tracheostomy [73]. The question of whether such a high number and different orientations depends exclusively on the patients or on societal effort for palliative care, or both, remains open.

We would therefore argue for the reformulation of the ‘extinction of thinking’ theory because the length of the period of complete immobility (with no chance to produce a desired effect in the environment) and the lack of rewarding contingencies might not be the only determinants of the extinction of goal-directed ideas and of the progressively reduced learning ability. Reduced skill learning in this condition is assumed to be dependent on the decline in synaptic plasticity and neuro-modulatory control [5].

Thus, the elements of the theory could then be assumed to take two forms, the physiological and the psychosocial one. The psychosocial determinant is intrinsically linked to the ethical and societal implications of communication with brain-based devices because they allow for quality-of-life assessments in completely locked-in persons and ensure the maintenance of their well-being. However, the psychosocial determinant will not be examined here because the factors and materials to be investigated are out of the scope of the present study.

## 5. Conclusions

We retrospectively analyzed the verbal communication of a completely locked-in person after the implantation of electrodes in the brain. The examination of the operant learning neurofeedback procedure and the sentences produced by the patient revealed preserved brain functioning at different levels and complementary domains, with the auditory system and language functional in all the examined domains. We argued that novel skill learning occurred in complete paralysis. The physical feedback and reinforcement in the stimulus–response–effect associative chains played an essential role within the instrumental (voluntary) learning procedure and for the internalization of ideas and strategies aimed at maintaining purposeful communication. The evidence proposed here calls for the reformulation of the ‘extinction of thinking’ theory, which should include the psychosocial determinant, in addition to the physiological one, for the extinction to take effect. The case presented here should fundamentally change the behavior of physicians, caregivers, and researchers. A so-called modern democratic society should accept that it is a “duty” to care for these persons and to do everything possible to provide them with the support to sustain communication, just like any other ill person. In case they communicate, good quality of life is secured most probably in the majority of these patients. If they cannot communicate, then a good quality of life has to be assumed. In short, we have the duty to recognize and respect their dignity, which in case of communication deficit implies providing supported measures to express needs, desires and thoughts, including the will to live.

## Figures and Tables

**Figure 1 sensors-24-06366-f001:**
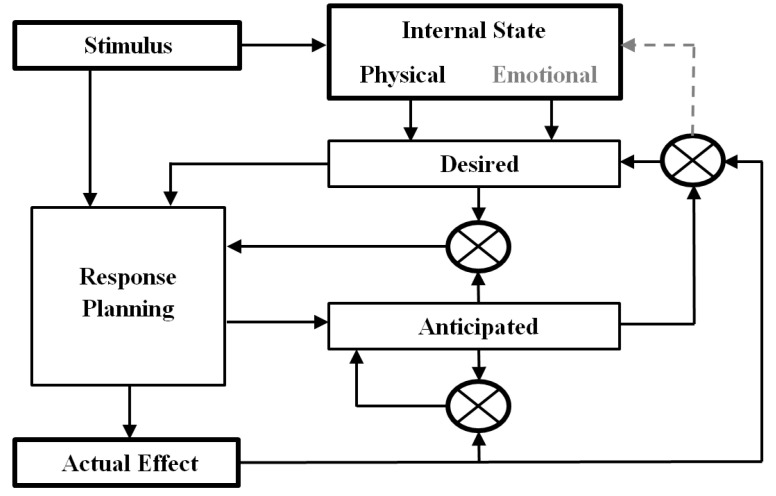
Learning model of the S–R–E unit. A simplified diagram of the stimulus–response–effect associative network, adapted from Ziessler et al. (2004) and Birbaumer et al. (2012) [4,5], is shown (licensed by Springer Nature, see Acknowledgments). The dashed line, which is proposed here as a change in the original model, indicates that the actual effect might have an impact on the “Internal State”.

**Figure 2 sensors-24-06366-f002:**
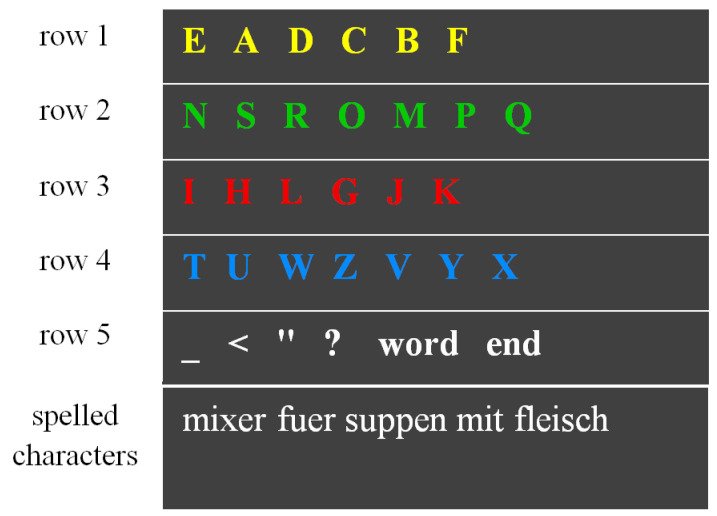
Graphical representation of the auditory speller used in the third and fourth communication periods, adapted from the Supplementary Video V2 (frame at 23 s) in Chaudhary et al. (2022) [14] (for the license of Supplementary Video V2 see Acknowledgments). The frame has been adapted for a 2D representation of the interface.

**Figure 3 sensors-24-06366-f003:**
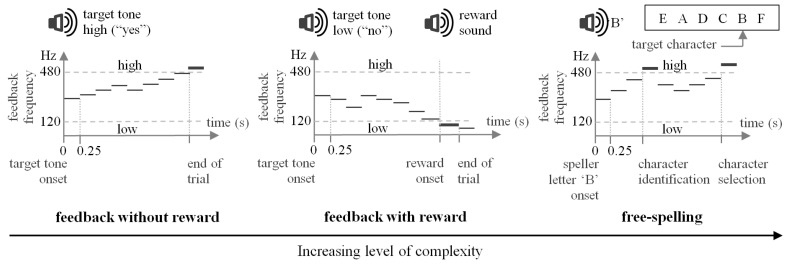
Representation of the gradual increase in complexity of the neurofeedback task (shaping). The three types of trial are qualitative examples of the neurofeedback mechanism. During free-spelling, the selection of the interface characters’ subsets and then of single characters (letter ‘B’ in this case) need to be accomplished twice, one for identification and one for confirmation.

**Table 1 sensors-24-06366-t001:** Communication strategies.

Number	Period	System/Task	Interface	Feedback Domain	Response-Effect Time Interval
1	from Aug. 2016 to Aug. 2017	Digital eye-tracker, eye movements controlling the pointer on the computer screen ^a^	MyTobii scheme (e.g., virtual keyboard)	visual	roughly < 70 ms
2	from Aug. 2017 to Feb. 2018	Manual spelling procedure, eye movements detected by observation to select characters based on “no”/“yes” responses	Transpaent board displaying colored alphanumeric characters	visual (auditory)	roughly < 10 s
3	from Feb. 2018 to Feb. 2019	Digital auditory speller, electro-oculogram of eye movements to select characters based on “no”/“yes” responses	Auditory scheme presenting alphanumeric characters	auditory	[3–10] s
4	from Jun. 2019 to Jun. 2020	Same speller as above (3) but used neuronal responses from neurons in SMA ^b^ to indicate “no” or “yes”, fed back as tones changing in frequency	Auditory scheme presenting alphanumeric characters	auditory (frequency range 120–480 Hz)	250 ms

^a^ During the calibration, discriminative stimuli are presented, and the user learns to control the pointer through a few S–R–E sequences (the delay between the eye movements and the movement of the pointer ranges within tens of milliseconds). ^b^ SMA is the supplementary motor area.

**Table 2 sensors-24-06366-t002:** Reported intelligible sentences.

Main Domain (Specific or Other Domains) ^a,b^	Deutsch	English	Day(s)
**Social**			
(feedback)	‘erst mal moechte ich mich niels und seine birbäumchen bedanken’	‘first I would like to thank Niels and his birbäumchen’	107
(instructions) ^c^	[sentence spelled in English]	‘turn on word recognition’	183
(wish) ^c^	[sentence spelled in English]	‘come tonight [to continue with the speller]’	203, 247, 251, 294, 295
(feedback) ^c^	[sentence spelled in English]	‘is it easy back once confirmation’	253
(instructions) ^c^	[sentence spelled in English]	‘tell alessandro i need to save edit and delete whole phrases and all of that into the list where (patient’s son name)’	295
(instructions, feedback) ^c^	[sentence spelled in English]	‘why cant you leave the system on. ifind that good’	461
(feedback)	‘jungs es funktioniert gerade so muehelos’	‘boys, it works so effortlessly’	247
(wish, drink)	‘und jetwzt ein bier’	‘and now a beer’ (through the gastro-tube)	247, 251, 253, 461
(wish, food)	‘mixer fuer suppen mit fleisch’	‘instructed his wife to buy a mixer for soup with meat’	247
(wish, food)	‘gulaschsuppe und dann erbsensuppe’	‘Gulash soup and sweet pea soup’	253
(wish, food)	‘wegen essen da wird ich erst mal des curry mit kartoffeln haben und dann bologna und dann gefuellte und dann kartoffeln suppe’	‘for food I want to have curry with potato then Bolognese and potato soup’	462
(emotion)	‘(son’s name) ich liebe meinen coolen (son’s name)’	‘I love my cool son’	251
(wish, emotion, vision)	‘(son’s name) willst du mit mir bald disneys robin hood anschauen’	‘Do you want to watch Disney’s Robin Hood with me’	253
(wish, emotion, vision)	‘alles von den dino ryders und brax autobahnund alle aufziehautos’	‘everything from dino riders and brax and cars’	309
(wish, emotion, vision)	‘(son’s name) moechtest du mit mir disneys die hexe und der zauberer anschauen auf amazon’	‘would you like to watch Disney’s witch and wizard with me on amazon’	461
**Memory**			
(wish, episodic memory)	‘wili ch tool balbum mal laut hoerenzn’	‘I would like to listen to the album by Tool [a band] loud’	245
(wish, social, somatosensory, episodic memory)	‘mein groesster wunsch ist eine neue bett und das ich morgen mitkommen darf zum grillen’	‘my biggest wish is a new bed and that tomorrow I come with you for barbecue’	462
**Sensory**			
(somatosensory)	‘kein shirt aber socken’	‘no shirts but socks [for the night]’	244
(emption)	‘mama kopfmassage’	‘Mom head massage’	247
(vision)	‘an alle muessen mir viel oefter gel augengel’	‘everybody must use gel on my eye more often’	254
**Proprioception**			
(instructions)	‘kop?f immerlqz gerad’	‘head always straight’	161
(instructions)	‘erstmal kopfteil viel viel hoeh ab jetzt imm’	‘first of all head position very high from now’	251
(instructions)	‘alle sollen meine haende direkten auf baubch’	‘everybody should put my hand direct on my stomach’	344
(instructions, social)	‘zum glotze und wenn besuchen da ist das kopfteil immer gaaanz rauf’	‘when visitors are here, head position always very high’	461

^a^ Language domain is functional to all the domains listed. ^b^ For the license of original data see Data Availability Statement. The list of sentences has been adapted for a domain-based representation. ^c^ Some sentences were spelled in English as they were directed to non-German native experimenters.

## Data Availability

The data presented in this study are available in the article by Chaudhary and colleagues (2022) at https://www.nature.com/articles/s41467-022-28859-8 (accessed on 5 August 2022), reference number [14] and are licensed under CC BY 4.0 (https://creativecommons.org/licenses/by/4.0/). Data presented in Table 2 can be found in the paragraph “Speller sessions” © in Chaudhary et al. (2022) [14], pp. 4–6. No new or additional data were collected in the present retrospective study.

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
