# Peer review of "Brain Function, Learning, and Role of Feedback in Complete Paralysis"

_sensors, 2024, doi:10.3390/s24196366_

Round 1

Reviewer 1 Report

Comments and Suggestions for Authors

This paper retrospectively analyzed the verbal communication of a completely paralyzed person after implantation of electrodes in the brain. Authors carried out a comparative examination of the communication results with current literature, and reformulate the hypothesis of extinction of response planning and goal-directed ideas in the completely locked-in state. There are still some issues that need to be addressed.

1. Authors need to briefly describe the data acquisition devices used in the last three periods.

2. It would be interesting to introduce the real-world performance such as reliability of the four types of communication interface.

3. Authors need to declare that the study was approved by the Ethics Committee.

Reviewer 2 Report

Comments and Suggestions for Authors

The authors report a locked-in state ALS patient who was implanted with Two 64 MEA in the left supplementary and primary motor cortex. By using the neurofeedback and reinforcement method and with the S-R-E model, the authors explore the prerequisites and determinants for learning to control a BCI for communication in paralysis. The presentation and arguments about ideomotor theory of thinking and the operant learning procedure for the reformulation of the previously postulated hypothesis of extinction of extinction of response planning and goal-directed ideas in this patient as an example.

The paper was submitted as a perspective article under the medical sensors section in the special issue of “Brain Computer Interface for Biomedical Applications”. The paper is heavily toward the brain cognitive function and communication responses to specific protocols. Neither the details of implantable BCI system nor the signal analysis with/without neurofeedback were covered in the paper for readers to understand how to quantify the selection of words to complete the sentence. I would not recommend publishing in the current format in this section.  

Reviewer 3 Report

Comments and Suggestions for Authors

Dear authors,

I present my suggestions and evaluation of your manuscript “Brain Function, Learning and Role of Feedback in Complete Paralysis” below.

The overall merit of the proposed manuscript is good. I am recommending accepting after minor revision. I am adding the main points and questions that need to be solved or clarified.

Questions and Quotes

  1. The ORCID should be added to all of the authors. 

  2. Please revise the citing literature, some of the newest references are missing. Only six references cited the research from the past three years. 

  3. Figure 3 is missing the scales and units.

  4. Table 2 - Why are some sentences only in the English version?
